# Modulation of Type I Interferon System by African Swine Fever Virus

**DOI:** 10.3390/pathogens9050361

**Published:** 2020-05-09

**Authors:** Elisabetta Razzuoli, Giulia Franzoni, Tania Carta, Susanna Zinellu, Massimo Amadori, Paola Modesto, Annalisa Oggiano

**Affiliations:** 1Department of Genoa, Istituto Zooprofilattico Sperimentale del Piemonte, Liguria e Valle D’Aosta, 16129 Genova, Italy; paola.modesto@izsto.it; 2Department of Animal Health, Istituto Zooprofilattico Sperimentale della Sardegna, 07100 Sassari, Italy; tcarta@uniss.it (T.C.); susanna.zinellu@izs-sardegna.it (S.Z.); annalisa.oggiano@izs-sardegna.it (A.O.); 3School of Veterinary Medicine, University of Sassari, 07100 Sassari, Italy; 4Laboratory of Cellular Immunology, Istituto Zooprofilattico della Lombardia e dell’Emilia Romagna (IZSLER), 25124 Brescia, Italy; massimo.amadori@izsler.it

**Keywords:** African swine fever, IFN-α subtypes, IFN-β, monocyte-derived macrophages, INF-α activation

## Abstract

African Swine Fever Virus (ASFV) has tropism for macrophages, which seems to play a crucial role in disease pathogenesis and viral dissemination. Previous studies showed that ASFV developed mechanisms to evade type I interferon (IFN) responses. Hence, we analyzed the ability of ASFV strains of diverse virulence to modulate IFN-β and IFN-α responses. Porcine monocyte-derived macrophages un-activated (moMΦ) or activated with IFN-α (moMΦ + FN-α) were infected with virulent (22653/14) or attenuated (NH/P68) ASFV strains, and expressions of IFN-β and of 17 IFN-α subtypes genes were monitored over time. ASFV strains of diverse virulence induced different panels of IFN genes: infection of moMΦ with either strains caused statistically significant up-regulation of IFN-α3, -α7/11, whereas only attenuated NH/P68 determined statistically significant up-regulation of IFN-α10, -α12, -α13, -α15, -α17, and IFN-β. Infection of activated moMΦ with either strains resulted in up-regulation of IFN-β and many IFN-α subtypes, but statistical significance was found only for IFN-α1, -α10, -α15, -α16, -α17 in response to NH/P68-infection only. These data revealed differences in type I IFNs expression patterns, with differences between strains of diverse virulence. In addition, virulent 22653/14 ASFV seems to have developed mechanisms to suppress the induction of several type I IFN genes.

## 1. Introduction

African Swine Fever virus (ASFV) is the etiological agent of African Swine Fever, a contagious viral disease of domestic pigs and wild boar; currently, it is present in many sub-Saharan African countries, Russian Federation, Trans-Caucasus, Eastern and Central Europe, and South East Asia [1,2]. The diffusion of this disease poses a threat to the swine industry worldwide, due to the lack of licensed vaccines or available treatments. Today, stamping out policies and movement restriction are the only measure useful to reduce the spread of the disease, causing important economic losses in the swine industry worldwide [3,4]. The etiological agent is a large, enveloped double-stranded DNA virus, belonging to the *Asfaviridae* family. ASFV has tropism for cells of the myeloid lineage, especially monocytes and macrophages, which seems to play a crucial role in disease pathogenesis, viral persistence and dissemination [1,2,3]. In vitro studies on macrophages showed that attenuated strains induce enhanced expression of key regulatory cytokines, like IL-12p40, TNF-α, and Type I Interferons (IFN-β and IFN-α1), and chemokines (CCL4, CXCL8, CXCL10) compared to high virulence strains [5,6,7,8,9]. In particular, recent studies demonstrated the ability of virulent ASFV to encode several genes that inhibit Type I IFN responses, i.e.; genes belonging to MGF 360, MGF 505, A276R, A528R, I329L regions [8,9,10]; deletion of some of these genes results in virulence attenuation [8,9,10,11]. Indeed, attenuated ASFV strains, such as NH/P68 and OUTR88/3, which are more sensitive to type I IFN-α [12,13], present loss or truncation of several MGF 360 and 505 genes compared to virulent ASFV strains [14].

IFNs are a family of proteins synthesized and secreted by different cell types. They were named after their capacity to interfere with viral infections [15]; however, they also show immunomodulatory and anti-proliferative activities, revealed in swine, as well [16]. The porcine IFNs system is very complex and among the three groups (I, II, III), porcine type I IFNs family is composed of at least 39 functional genes, including one IFN-β, αω, -ε, and -κ, 17 IFN-α, 11 IFN-δ, and 7 IFN-ω subtypes [17]. In pigs, IFN-β is encoded by one gene only, whereas IFN-α is a multigene family with 17 functional genes [17]; this gene family shares high identity at both nucleotide (96–99.8%) and amino acid level (91.1–100%) [18]. Despite their high structural homology, porcine IFN-α subtypes exhibit different anti-inflammatory, MHC modulation, and antiviral activities against several pathogens [19,20,21], thus differences in the production of IFN-α subtypes might influence intensity and duration of an antiviral response. Accordingly, our working hypothesis implied that virulent isolates might have developed mechanisms to suppress selected type I IFN types and subtypes in their target cells; attenuated strains might have lost, at least partially, some of these peculiar mechanisms.

The knowledge of this branch of innate immunity would improve our understanding of the early stage of ASFV pathogenesis and might aid the rational design of ASFV vaccines. In addition, a better understanding of ASFV-driven modulation of type I IFN system might help design antiviral agents or a metaphylactic intervention strategy against this virus. In fact, it has been reported that adenovirus-mediated type I interferon expression delayed appearance and reduced clinical signs in pigs infected with another virus, such as classical swine fever virus (CSFV) [20], and completely protected pigs from foot and mouth disease virus (FMDV) [21]; hence, a similar strategy might be adopted against ASFV, too.

In this conceptual framework, the aim of our study was to investigate the pattern of expression of different type I IFN (IFN-β, 17 IFN-α subtypes) in response to ASFV infection. We compared an attenuated (NH/P68) and a virulent (22653/14) strains, in order to investigate how the level of attenuation affects the virus ability to induce type I IFN responses.

## 2. Results

### 2.1. Evaluation of IFNs Gene Expression on moMΦ

moMΦ were infected with either attenuated NH/P68 or virulent 22653/14 ASFV strains using multiplicity of infection (MOI) 1; in our previous work, intracellular levels of late viral proteins p72 and virus infectious particles in culture supernatants were determined at 21 h pi [13]. We observed that using this MOI 40–60% of ASFV-infected moMΦ presented late ASFV protein p72 intracellularly at 21 h pi [13]. In this work, IFNs gene expression was monitored over time (3, 6, 9, 12, 21 h pi). Infection with NH/P68 caused a significant increase of IFN-β gene expression in moMΦ at different times pi; in particular, we observed up-regulation at 6 (*p* < 0.05), 9 (*p* < 0.05), 12 (*p* < 0.05), 21 (*p* < 0.001) hours post-infection with respect to time 0. On the contrary, the virulent 22653/14 strain did not modulate IFN-β gene expression with statistical significance (Figure 1). Concerning the IFN-α subtypes, we observed a statistically significance increase of IFN-α3 and IFN-α7/11 gene expression after infection with either NH/P68 or 22653/14 (*p* < 0.05) (Figure 1). Increased gene expression was also observed for IFN-α9, but was not statistically significant. NH/P68 caused a broader IFN-α response compared to the virulent 22653/14, with statistically significant induction of IFN-α10 (*p* < 0.05), α12 (*p* < 0.05), -α13 (*p* < 0.05), -α15 (*p* < 0.05), -α16 (*p* < 0.01), -α17 (*p* < 0.05) 24 h pi (Figure 1). The other genes under study were not significantly (*p* > 0.05) modulated by this virus, although an increase of IFN-α5/6, -α8, -α14 for both strains was observed.

### 2.2. Evaluation of IFNs Gene Expression on moMΦ + IFN-α

IFN-α activated moMΦ were infected with either attenuated NH/P68 or virulent 22653/14 ASFV strains using MOI 1. In our previous work, we observed that using this MOI about 40–50% of ASFV-infected moMΦ activated with IFN-α were infected at 21 h pi [13]. IFNs gene expression was also monitored over time in activated moMΦ. NH/P68 infection of moMΦ activated with IFN-α caused a significant (*p* < 0.01) increase of IFN-β gene expression; in particular, we observed up-regulation, with respect to time 0, at all time points with the exception of 3 h post-infection. In addition, the virulent 22653/14 induced moderate IFN-β induction, with statistical significance at 9 h pi (*p* < 0.05) (Figure 2). Regarding IFN-α, almost all subtypes were induced, but without statistical significance for IFN-α3 and -α7/11 (*p* > 0.05). Both strains induced IFN-α2, -α4, -α5/6, -α8, -α9, -α12, -α13, -α14 induction, but differences were observed between strains: NH/P68 up-regulated -α5/6, -α8, -α12, -α13, -α14 at more times pi compared to virulent 22653/14 (Figure 2). NH/P68 only up-regulated gene expression of IFN-α1 at 9 (*p* < 0.05) and 12 (*p* < 0.01) h pi, IFN-α10 at 9 (*p* < 0.05), 12 (*p* < 0.01) and 21 (*p* < 0.05) h pi, and of -α15, -α16, -α17 at 12 h pi (*p* < 0.05) (Figure 2).

### 2.3. Evaluation of IFNs Release after Virus Infection

Our ELISA results did not show significant differences (*p* < 0.05) in IFN-β and IFN-α1 release after infection with both attenuated or virulent ASFV strains (Figure 3). The same trend was observed for unactivated or IFN-α activated moMΦ (Figure 3). Differences between strains mainly concerned IFN-β levels in moMΦ + IFN-α: following 22653/14 infection, secreted levels were lower compared with infection by attenuated NH/P68 strain, although differences were not statistically significant (*p* = 0.083, tendency).

## 3. Discussion

Type I IFNs are a heterogeneous group, composed of distinct families (IFN-α, IFN-β, IFN-ε, IFN-ω, IFN-κ, IFN-δ and IFN-τ), with some of them (like IFN-α) consisting of different subtypes [17]. In pigs there are 17 IFN-α subtypes, which present different antiviral activities against pseudorabies virus (PRV), classical swine fever virus (CSFV), porcine reproductive and respiratory virus (PRRSV), and vesicular stomatitis virus (VSV) [17,19,20]. Therefore, differences in their production by infected cells might influence the quality and the extent of an antiviral response. It can be speculated that virulent AFSV might have developed immune escape mechanisms based on selective inhibition of some IFN-α subtypes.

We observed higher induction of IFN-α1 after infection with the attenuated NH/P68 compared to the virulent 22653/14, with statistical significance in moMΦ + IFN-α, and these findings are in accordance with Gil et al. (2008), where researches observed higher induction of IFN-α1 in macrophages after infection with NH/P68 compared to virulent L60 [6]. IFN-α1 presents strong antiviral activity against PRV [19], PRRSV, and VSV [17], and we recently described that ASFV strain of diverse virulence presented different sensitivity to IFN-α1 antiviral activity: 100 U/mL of IFN-α1 inhibited NH/P68, but not 22653/14, in moMΦ, as assessed by reduction of viral levels in culture supernatants [13]. Our results and those of Gil et al. [6] suggest that virulent 22653/14 might have developed mechanisms to suppress induction of this IFN-α subtype, and these are at least partially lost in attenuated NH/P68.

NH/P68 infection of moMΦ resulted in up-regulation of other IFN genes compared to virulent 22653/14, with statistical significance for IFN-α10, -α12, -α13, -α15, -α16, -α17. Nevertheless, both strains significantly induced IFN-α3 and IFN-α7/11 expression. Interestingly, it was reported that porcine recombinant IFN-α3, -α7, -α11 displayed no antiviral activity against PRV [19] and both IFN-α7 and -α11 presented no antiviral activity against PRRSV and VSV [17]. In addition, Sang et al. (2010) observed that PRRSV antiviral activity positively correlated with induction of MxA: all recombinant IFN-α subtypes induced expression of this interferon stimulated gene, with the exception of IFN-α7 and IFN-α11 [17], and the MxA protein can also severely impair ASFV replication [22].

About activated moMΦ + IFN-α, a broader induction of IFN-α subtypes by both strains was observed, but still stronger after infection with NH/P68 compared to 22653/14 ASFV. The virulent Sardinian isolate seems to have developed mechanisms to suppress innate immune responses. NH/P68 only induced gene expression of IFN-α1, -α10, -α15, -α16, -α17 in activated moMΦ. Interestingly, it was described that IFN-α15 and IFN-α16 downregulated expression of MHC class I [19], and we observed that infection attenuated NH/P68, but not virulent 22653/14, resulted in MHC I downregulation in either moMΦ or moDC [13,14,15,16,17,18,19,20,21,22,23]. This is likely to provide a crucial advantage to the virulent ASFV strain in terms of an outright escape strategy from surveillance of NK cells based on “missing self” [24]. In addition, future studies will hopefully investigate similar mechanisms related to the expression of stress antigens, like MHC class I-related molecules A and B (MICA and MICB) and the family of UL16-binding proteins (ULBP1-6), recognized by NKG2D of NK and NKT cells [24].

Nevertheless, information regarding biological activities of diverse IFN-α subtypes are still limited, and future studies should be performed in order to better understand how different IFNs expression pattern can affect host antiviral response. In vitro experiments, like viral yield reduction assays, will help understand the impact of these molecules on ASFV replication. In addition to that, in vivo studies are badly needed to properly characterize the immunomodulatory activities of these molecules, thus revealing the real extent of their protective effects during ASFV infection.

Concerning IFN-β, we observed that NH/P68 infection induced higher expression of this gene compared to the virulent 22653/14, in accordance with both Reis et al. (2016) (OURT 88/3 vs. Benin 97/1) and Garcia-Belmonte et al. (2019) (NH/P68 vs. Armenia07) [8,9]. As previously stated, NH/P68 and OUTR88/3 present loss or truncation of several genes within MGF 360 and 505 regions [14], which seem to play a crucial role in suppressing IFN-β gene expression [8].

Although our data showed that the attenuated NH/P68 induced higher induction of type I IFN compared to the virulent 22653/14, we could not detect higher production of either IFN-α1 or IFN-β from infected moMΦ, in accordance with a previous study [25]. The authors do not rule out the possible release of minute amounts of Type I IFNs, not revealed by commercial ELISAs and even beneath the usual detection levels of antiviral assays in tissue cultures. The absence of increased type I IFN production by NH/P68-infected macrophages is likely due to post-transcriptional mechanisms, which could involve either mRNA stability, alternative splicing or translation [26]. Nevertheless, lower levels of IFN-β were observed in 22653/14-infected compared to mock-infected or NH/P68-infected activated moMΦ. This implies an outright inhibition of IFN-β as a result of 22653/14 infection. Differences were not statistically significant, but this statistical tendency was similar to what recently described by Garcia-Belmonte et al. in porcine alveolar macrophages. It was observed that virulent Arm07 ASFV induced IFN-β levels lower than those of uninfected controls [9], suggesting that virulent ASFV strains actively suppress the host’s immune responses.

Overall, our results highlight that the attenuated NH/P68 induced a stronger type I IFN response compared to virulent 22653/14, with statistically significant induction of IFN-β and several IFN-α subtypes in moMΦ. Nevertheless, expression of some IFN-α subtypes (-α3 and -α7/11) in moMΦ was induced by both strains. Regarding activated moMΦ, both IFN-β and diverse IFN-α subtypes were induced by ASFV, but selectively (IFN-α1, -α10, -α15, -α16, -α17) or with stronger intensity (IFN-β, IFN-α5/6, -α8, -α12, -α13, -α14) by NH/P68 compared to the virulent Sardinian isolate. Overall, virulent 22653/14 seems to have developed mechanisms to suppress the induction of several type I IFN genes. In addition, we observed that IFN-α subtypes were differently modulated upon infection, highlighting the importance of future studies to better define the biological properties of these subtypes, and to understand how differences in their expression pattern can affect intensity and duration of host antiviral responses. Data generated in this study could contribute to better understand ASFV-driven modulation of type I IFN system, to improve rational design of vaccines or antiviral agents.

## 4. Materials and Methods

### 4.1. Animals

Whole blood was obtained from healthy, 6- to 24-month-old, cross-bred pigs (*Sus scrofa domesticus*), housed at the Experiment Station of IZS of Sardinia (Sassari, Italy). The local ethical committee approved all procedures performed, in agreement with the Guide of Use of Laboratory Animals issued by the Italian Ministry of Health. The ASFV-negative status was assessed by real time PCR (EDTA blood samples), ELISA (Ingezim PPA Compac^®^, Ingenasa, Madrid, Spain) and Immunoblotting test (serum samples), as described in the Manual of Diagnostic Tests and Vaccines for Terrestrial Animals [27].

### 4.2. Viruses

Two ASFV strains of diverse virulence were used in this study: the virulent hemoadsorbing 22653/14 (isolated from a naturally infected pigs during an ASF outbreak in Sardinia in 2014) (Exotic Disease Laboratory ASF Virus Archive, IZS of Sardinia, Sassari, Italy) and the attenuated NH/P68 (kindly provided by the EU ASF Reference Laboratory CISA-INIA, Madrid, Spain). NH/P68 is non-hemoadsorbing and was isolated in 1968 from a chronically infected pig in Portugal [14,15,16,17,18,19,20,21,22,23,24,25,26,27,28]. Both ASFV strains belong to genotype I [29,30]. ASFV strains were propagated in vitro using 25 cm^2^ flask (Corning, Corning, NY, USA), by inoculation in of sub-confluent monolayers of porcine monocytes/macrophages, as previously described [27,28,29,30,31]. Mock-infected controls were prepared in an identical manner from uninfected monocyte/macrophage cultures. Viral titers were determined by 10-fold serial dilutions of viral stocks on monocyte/macrophages in 96-well plates, followed by immunofluorescence staining five days post infection [27]. Viral titers were then calculated using the Spearman–Kärber formula.

### 4.3. Cells

Porcine leukocytes were obtained from whole blood and heparin was used as an anticoagulant. Macrophages were generated in Petri dishes using human M-CSF (hM-CSF), as previously described [13,14,15,16,17,18,19,20,21,22,23,24,25,26,27,28,29,30,31,32]. Briefly, leucocytes were cultured at 37 °C, 5% CO_2_ in Petri dishes in cRPMI (RMPI, 10% fetal bovine serum (FBS), 100 U/mL penicillin and 100 μg/mL streptomycin) supplemented with 50 ng/mL of recombinant hM-CSF (Thermo Fisher Scientific, Waltham, MA, USA) for seven days. Then non-adherent leukocytes were removed, adherent cells were detached by gentle scraping with a pipette, centrifuged at 200× *g* for 8 min, and number of viable cells were determined using a Countess Automated Cell Counter (Thermo Fisher Scientific). 5 × 10^5^ live cells/mL were seeded in a 12 well plates (Greiner CELLSTAR, Sigma-Aldrich, Saint Louis, MO, USA) in cRPMI (1.5 mL/well). After seeding, macrophages were cultured at 37 °C 5% CO_2_ for another 24 h: they were left untreated (moMΦ) or stimulated with 100 U/mL recombinant porcine IFN-α (PBL Assay Science, Piscataway, NJ, USA) (moMΦ + IFN-α).

### 4.4. ASFV Infection of Macrophage

moMΦ or moMΦ + IFN-α were infected with virulent 22653/14 or attenuated NH/P68 using a multiplicity of infection (MOI) of 1. After 90 min at 37 °C 5% CO_2_ (adsorption period), viral inoculum was removed, cells were washed with medium and fresh cRPMI was added to the wells [13]. Cells were incubated at 37 °C 5% CO_2_ and harvested after 3, 6, 9, 12, 21 h post-infection (pi), spanning the approximate life cycle of ASFV replication [8,33] At each time points, supernatants were removed, and cells were stored at −80 °C until analyzed. Experiments were performed in technical duplicate (2 wells each condition) and repeated six times using different blood donor pigs. At 21 h pi, supernatants were also collected, clarified by centrifugation at 2000× *g* for 3 min and stored at −80 °C until analyzed [13].

### 4.5. Evaluation of IFNs Gene Expression

Gene expression of IFN-β and 17 different IFN-α subtypes in ASFV-infected moMΦ at selected time-points (0, 3, 6, 9, 12, 21 h pi) was determined by RT qPCR using primer sets previously described [34,35]. Briefly, in each sample total RNA was extracted using RNeasy Mini Kit (Qiagen, Hilden, Germany) in agreement with the manufacturer’s instructions. Then, 100 ng of purified RNA were used as template for cDNA synthesis; EVA Green Real-Time PCR amplification was performed in a CFX96™ Real-Time System after the reverse transcription step, as previously described [16]. In each sample, the relative expression of the selected genes was calculated using the formula 2-ΔΔCt where:ΔΔCt = ΔCt (mock) − ΔCt (target gene after infection).

After calculation of 2-ΔΔCt and the Kolmogorov-Smirnov test for normality, data sets were checked for statistically significant differences. We performed six experiments using different blood donor pigs.

### 4.6. Analysis of Ttype I IFN Levels in Culture Supernatants

IFN-α1 and IFN-β levels in culture supernatants were measured by a Swine IFN-α Do-it-Yourself ELISA kit (King Fisher Biotech, St Paul, MN, USA) or a porcine IFN-β ELISA kit (MyBiosource, San Diego, CA, USA), respectively, according to manufacturer’s directions. An Epoch microplate reader (BioTek, Winooski, VT, USA) was used to read adsorbance. Experiments were performed in technical duplicate and repeated three (IFN-α) or four (IFN-β) times using different blood donor pigs.

### 4.7. Data Analysis and Statistics

Experiments were performed in technical duplicate, with three/four (ELISA) or six (qPCR) biological replicates using different pis as the source of moMΦ. Data are presented as mean with standard deviations (SD). GraphPad Prism v8.01 (GraphPad Software Inc, La Jolla, USA) and Minitab (version of 2019) (Minitab Inc.; Coventry, UK) were used to perform graphical and statistical analysis. All data were checked for normality using the Anderson-Darling test; virus effects on IFNs expression and production were analyzed by the parametric one-way ANOVA or the non-parametric Kruskal-Wallis followed by Dunn’s multiple comparison test. The significance threshold was set *p* < 0.05. A tendency was declared at *p* < 0.1.

## Figures and Tables

**Figure 1 pathogens-09-00361-f001:**
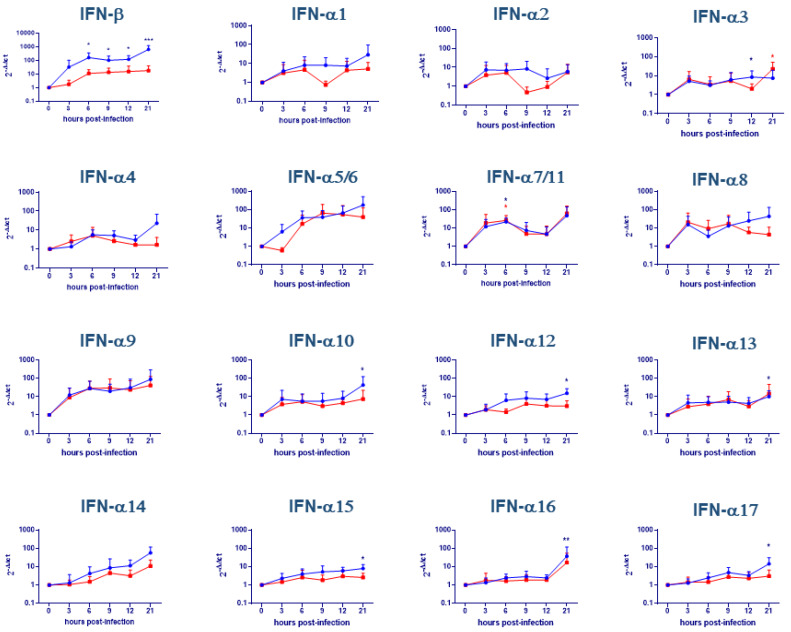
Cytokine gene expression in African Swine Fever Virus (ASFV)-infected macrophages. moMΦ were infected with the low virulence NH/P68 (blue) or the virulent 22653/14 (red) strains using an MOI 1. At 0, 3, 6, 9, 12, 21 h pi, gene expression of IFN-β and 17 IFN-α subtypes were determined using qPCR. Data were normalized on the values of un-infected control (0 h pi) and expressed as ΔΔCt, where ΔΔCt  =  (ΔCt observed in un-infected moMΦ) − (ΔCt observed in ASFV-infected moMΦ). The mean data + SD from six independent experiments utilizing different animals are shown. Values of ASFV-infected cells were compared to the corresponding mock-infected control using a one-way ANOVA (IFN-α12, IFN-α17) or a Kruskal-Wallis test (all the others), followed by Dunn’s multiple comparison test; *** *p* < 0.001, ** *p* < 0.01, * *p* < 0.05.

**Figure 2 pathogens-09-00361-f002:**
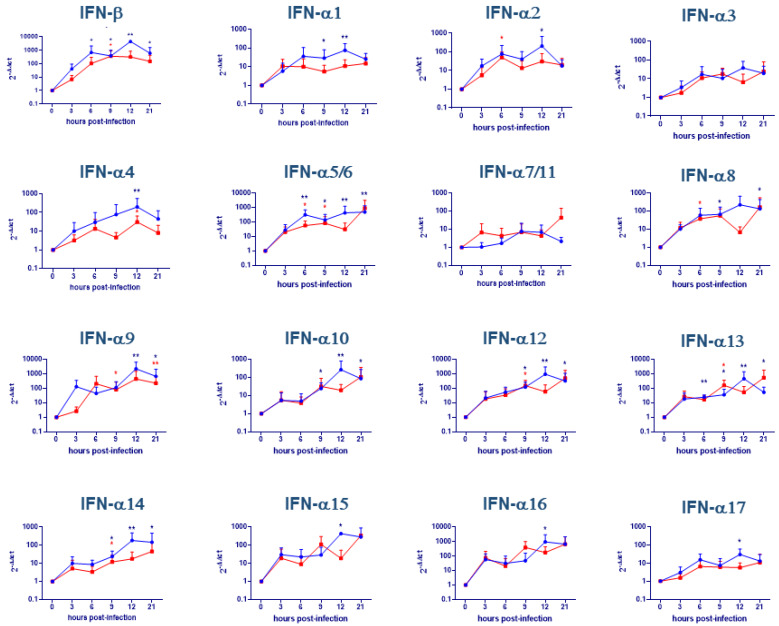
Cytokine gene expression in ASFV-infected IFN-α macrophages. moMΦ were activated for 24 h with IFN-α and then infected with the low virulence NH/P68 (blue) or the virulent 22653/14 (red) strains using an MOI 1. At 0, 3, 6, 9, 12, 21 h pi, gene expression of IFN-β and 17 IFN-α subtypes were determined using qPCR. Data were normalized on the values of un-infected control (0 h pi), and expressed as ΔΔCt, where ΔΔCt  =  (ΔCt observed in un-infected moMΦ) − (ΔCt observed in ASFV-infected moMΦ). The mean data + SD from six independent experiments utilizing different animals are shown. Values of ASFV-infected cells were compared to the corresponding mock-infected control using a Kruskal-Wallis test, followed by Dunn’s multiple comparison test; *** *p* < 0.001, ** *p* < 0.01, * *p* < 0.05.

**Figure 3 pathogens-09-00361-f003:**
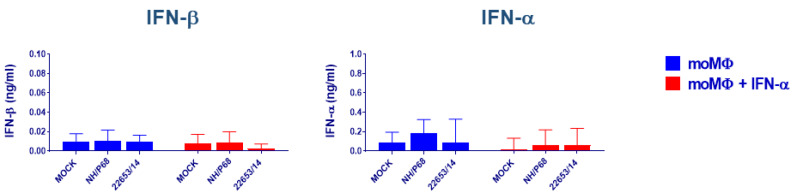
IFN-β and IFN-α release by macrophage in response to infection with ASFV strains of diverse virulence. moMΦ were left untreated or activated with IFN-α. 24 h post-activation, supernatants were removed, then cells were infected with the low virulence NH/P68 or the virulent 22653/14 strains using an MOI 1, alongside mock-infected controls. At 21 h pi, the amount of IFN-β, IFN-α1 in culture supernatants were determined using commercial ELISA. The mean data + SD from four (IFN-β) or three (IFN-α) independent experiments utilizing different animals are shown. Values of ASFV-infected cells were compared to the corresponding mock-infected control using a Kruskal-Wallis test (IFN-α moMΦ) or a one-way ANOVA followed by Dunn’s multiple comparison test; *** *p* < 0.001, ** *p* < 0.01, * *p* < 0.05.

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
