# Peer review of "Modulation of Type I Interferon System by African Swine Fever Virus"

_pathogens, 2020, doi:10.3390/pathogens9050361_

Round 1
Reviewer 1 Report
In this manuscript, the Authors describe the modulation of type I IFN (17 subunits of IFN I -α, and IFN I –β) known as antiviral factor in host organism. Comparison of cell response (swine macrophages - target cells for ASFV) to infection of two different ASFV strains, revealed not surprising but very interesting and valuable results that low virulent strain generally cannot downregulate expression of genes responsible for antiviral activity.
Lack of effective vaccines against ASFV proves of complicated immune response to the pathogen. In this light, such studies as given by Authors are opportunity to close up on immunology and pathogenesis of ASFV infection.
Manuscript is well organised. The authors give an overall good background and discussion. Methods are sound. Work presents novelty.
Only few minor comments to the Authors:
- Please consider to move few sentences about general role and origin of IFNs from discussion to background section, thus may improve understanding of paper.
- Abstract section – line 20: INF instead IFN
- Keywords section – line 31: the same issue
- Introduction section – line 42, recent studies suggest ASFV is no longer only member of Asfaviridae (1)
- M&M section – line 203: Sus scrofa should be use as latin name of wild boar, for pig please use Sus scrofa domesticus, please write latin names in italics
I’ve got no further comments work done by Authors is impressive.
Reviewer 2 Report
The study discusses gene Expression of macrophage cells exposed to a virulent and an attenuated strain of African Swine Fever. Overall, this is a gene Expression study it does not show functionality of virus being suppressed in term of reduced viral loads in cell sthat had some high expression of some inIFN genes.
Since this is purely a gene expression study, the conclusion in Line 29-30 that virulent 22653/14 ASFV seems to have developed mechanisms to suppress the host’s antiviral response does not hold because there was no study showing reduction or increase or decrease in viral loads impacted by the genes expressed to warrant this conclusion. Similarly, lines 195-196, the conclusion does not hold because this is just expression of IFN genes of which the difference between the two Groups has to be supported With functional studies to prove that there is suppression of innate immune responses. mRNA Expression Levels do not always support results from functional studies.
It would be good for the Authors to discussion how the individual genes expressed impact on virus growth using examples where the same virus was shown to be suppressed by the genes used in this study. Authors should also discuss why the other genes that do not show differences between the two Groups might not have an impact on replication of avirulent and virulent strains of ASFV. In the Current state they have just Grouped the genes as one batch, which is significantly different from the other from cells treated with the avirulent and virulent strains ASF strains. You need to do some comparison and contrasting the functional activites of the selected genes in relation to ASF viral virulence.
Why didn't the Authors do a virus quantification analysis in relation to gene Expression? this would have helped determine the suppressive effect of IFN in reducing viral loads.
I seem not find the number of replicates used for each gene analyzed and whether the study was only done once. How did the Authors Ensure Equal input of template, cells used, etc? How many 12 well plates were used? or how many replicates of each time point were sampled? Why was the study ended at 21 hours post infection and not further? please explain why the chosen sampling timepoints were selected and not any other.
Round 2
Reviewer 2 Report
The manuscript is a second submission after responding to earlier Queries. In the Current submission Authors have responded to all Queries. The manuscript is pured based on gene Expression and in the Current submission it has been refined to discuss only on gene Expression.